# Performance Analysis of Additively Manufactured Hydraulic Check Valves with Different Postprocessing

**DOI:** 10.3390/ma16237302

**Published:** 2023-11-24

**Authors:** Agnieszka Klimek, Janusz Kluczyński, Jakub Łuszczek

**Affiliations:** Institute of Robots & Machine Design, Faculty of Mechanical Engineering, Military University of Technology, Gen. S. Kaliskiego 2 St., 00-908 Warsaw, Poland; agnieszka.klimek@wat.edu.pl (A.K.); jakub.luszczek@wat.edu.pl (J.Ł.)

**Keywords:** additive manufacturing, M300 steel, hydraulic check valves, internal leaking, surface analysis

## Abstract

Due to the need to use very precise manufacturing processes, hydraulic applications are one of the most demanding parts in production. Such a feature requires using molded and properly machined parts. On the other hand, such an approach makes hydraulic parts very heavy and requires the use of large amounts of material. One of the most promising manufacturing technologies that could be a real alternative to hydraulic parts production is additive manufacturing (AM). This paper aims to study how the AM process affects the performance properties of the as-built state, and investigate changes after different types of postprocessing in the case of hydraulic check valves. Based on the obtained results, using proper postprocessing is a crucial feature of obtaining check valves that perform their functions in a hydraulic system. In as-built parts, the surface roughness of the valve seats significantly exceeds the acceptable range (almost nine times—from 4.01 µm to 33.92 µm). The influence of the surface roughness of the valve seats was verified via opening pressure and internal leakage tests based on ISO standards. The opening pressures in all tested samples were similar to those in the conventionally made counterparts, but in the case of internal leakage only a fully finished AM valve revealed promising results. The obtained results could be useful for various enterprises that are seeking weight reduction possibilities for their low-volume manufactured products.

## 1. Introduction

Downsizing (in terms of mass and size) is the most important trend in modern manufacturing, as it results in decreased fuel consumption and lower pollutant emissions. What is more, with the reduced weight, it is possible to implement hydraulic systems in branches where the mass of the machine is the limitation, like unmanned aerial vehicles (UAV). Currently, the most popular kind of drive is electric, where the battery is a source of power. Due to that fact, the range, time of operation, and capacity are limited. There is some research on UAVs with a manipulator, but they can grasp only lightweight objects and have a small workspace [1,2,3]. What is more, manipulator mass is a crucial parameter in terms of UAV stability, as it strongly affects the location of the center of gravity.

One of the attempts to improve the capability of manipulation was presented by Lin Tianyu et al. in their article [4]. They decided to implement a hydraulic manipulator in a UAV. It ensures a significant improvement in torque and force, with the same power, and simplifies mechanical structure. Flowcopter Inc. went even further, as they proposed a hydraulic-driven quadrocopter. A hydraulic digital displacement pump (DDP) is driven by an engine and distributes hydraulic oil to hydraulic motors placed in rotors. DDP is a crucial component of the whole system and consists of a traditional piston unit with an independent outlet and electrically controlled check valves. Control of the check valves ensures proper performance of the whole unit, so it is extremely important to reduce the mass of hydraulic components. The way to miniaturize the check valve was presented by Pendzialek et al. [5]. They replaced the traditional body of the valve with a compliant mechanism, which serves as the spring and the closing element at the same time. The mass and size of the valve were improved, but the leakage was not investigated. Such a parameter is very important due to the performance properties related to each application [6]. Thus, there is a need to improve the dimension and weight of the valve without significant losses in performance. The way to achieve this goal may be through additive manufacturing (AM). During the significant growth of the usage of AM technologies over the last two decades, the main application of this technology was in moving through the prototyping, tooling, and low-scale production of final products. At the same time, scientific interest in that kind of technology was equally raised [7,8]. Nowadays, there is more and more research work related to specific case studies. One of the most interesting use cases of AM technologies is related to hydraulic systems [9] that are characterized by very good performance properties, but whose usage at the same time significantly increases the weight of the whole device. Such a factor is especially important in aircraft applications, where every gram of the total weight is important. That is why it is crucial to fit the possibilities of AM technologies for the production of parts for hydraulic systems [10,11,12,13]. The possibilities of using AM to reduce the mass of the produced parts are strictly related to the usage of lightweight structures [14] made of expensive materials [15]. What is more, the performance properties of parts produced via AM technologies could possibly be improved with the use of different postprocessing [16,17,18]. One of the heaviest parts of hydraulic parts are housings, which is why much research is related to the mass reduction of that kind of part. A good example is the research of Kneissl et al. [19], who used topological optimization to reduce the weight of the hydraulic housing for a brake-by-wire system in electrical drives. With regard to the maintenance of the functional properties of the housing, the authors were able to reduce the weight of the housing by 13% in comparison to the conventionally made counterpart. Significantly better results were obtained by Diegel et al. [20], who used a more advanced method called design for additive manufacturing (DfAM). The optimization that they used allowed them to improve the functionality of the hydraulic manifold, and at the same time produced weight savings of over 91% compared to the original manifold. An additional advantage of using the DfAM method was revealed during the support material optimization, where the authors obtained a 2% total material volume saving in comparison to the total volume of material in the part. Another approach was made by Li et al. [21], who analysed the pressure loss in the AM hydraulic flow channels and created a finite element model (FEM) characterized by a 7.72% error in comparison to the validation results. It is worth highlighting that, during the research, the authors used electrochemical polishing to reduce surface roughness values that could affect the pressure loss values. In the case of hydraulic parts, there is also a significant issue related to heat generation in the whole hydraulic system. To analyze this issue a quite different point of view was taken into account by Göltaş et al. [22], who analysed the thermo-hydraulic performance in a heat exchanger. Along with the conducted research, 23% more effectiveness was achieved in accordance with the reference part. At the same time, it was registered that using AM-made parts affects growth–flow turbulences, which was explained by the authors as being related to the high surface roughness of as-built parts. Using AM technologies for high-volume production is unjustified in comparison to conventional methods because of the significant cost of such kinds of manufacturing (i.e., metallic powder and amortization of the machine purchase cost). The sample purchase cost of the primary valve is about EUR 25, while the AM production cost of the same part is about EUR 400. This is why the usage of AM methods is appropriate for the unit production of parts characterized by complex geometries (i.e., lattice structures) and in emergency situations (i.e., battle-damage repair (BDR) in military applications, or the lack of availability of parts due to broken supply chains). Based on this review of the present state of the field, there is a visible and significant lack in the research into the typical hydraulic parameters of the parts obtained via AM technology. That is why the main aim of this research is to analyze the basic flow properties of the AM check valves and to compare the obtained results to conventionally made commercial parts. Such an analysis would be a helpful tool for DfAM regarding that kind of part and would answer the very important question regarding how advanced postprocessing should be provided to obtain the proper performance properties of hydraulic check valves.

## 2. Materials and Methods

### 2.1. Research Procedure

Research based on elements of additive manufacturing requires many additional activities related to the production process. In order to clearly present the research procedure with its emphasis on additive manufacturing, it is demonstrated in Figure 1. A more detailed description is presented in individual sections.

### 2.2. Material

Gas-atomized M300 maraging steel (Carpenter Additive, Philadelphia, PA, USA) was employed in AM of all sample parts. The powder particles had a typical spherical form (with diameters ranging from 20 to 63 µm), and in a minority of cases displayed satellites on the outer surface.

### 2.3. AM Process Description

Samples were manufactured using an SLM 125HL AM system (SLM Solutions, Lubeck, Germany) equipped with a 400 W laser source. The geometry of the simple sample was created by means of SolidWorks 2022–2023 software based on the microscopical measurements of a commercial check valve, RV-08, manufactured by Stauff (Stauff, Werdohl, Germany) (Figure 2). According to the data sheet [23], the opening pressure is 0.05 MPa and the maximum operating pressure is 50 MPa.

During the design process, five types of work pieces were prepared: Check valve with all dimensions kept (including threads)—in further description, named A.Check valve with all dimensions kept (excluding threads—for additional postprocessing of threads), named B.Check valve with external dimensions kept (for additional postprocessing of internal dimensions and threads), named C.Check valve with all dimensions kept (including threads), lattice structure, and 1 mm of wall thickness, named D.Check valve with all dimensions kept (including threads), lattice structure, and 2 mm of wall thickness, named E.

For a better understanding of the suggested iterations of the check valves, an additional structural diagram was made and shown in Figure 3.

There were two main reasons for the selection of the abovementioned check valve designs: Weight reduction while maintaining the external shape of the parts (D and E).Simplifying the postprocessing procedure—which would shorten the total manufacturing time (A, B, and C).

All of the designed geometries were preprocessed via Magics (Materialise, Leuven, Belgium, version 19), and the AM process was performed under standard conditions via laser powder bed fusion (PBF–LB/M). This was conducted within an argon atmosphere, with an oxygen content of less than 0.1%. The following process parameters (shown in Table 1) were utilized for sample production:

After the process, all parts were subjected to the typical postprocessing methods that are recommended by AM system producers: additional sandblasting of external surfaces and, for samples 1, 4, and 5, the finishing and tapping of the AM threads. Samples 3 and 4 were postprocessed via the additional machining of the threads (samples 4, and 5) and internal geometry (sample 5). All the produced samples with their assigned numbers are shown in Figure 4.

### 2.4. Microscopical Analysis

For optical dimensions measurements and surface roughness analysis, a Keyence VHX 7000 optical microscope (Keyence International, Osaka, Japan) was used. The roughness measurements were made in accordance with the following standard: PN-EN ISO 21920-2:2022-06 [24]. The area that was subjected to surface roughness measurement (Ra) is the conical surface of the seat that interacts with the poppet. Due to its geometric dimensions, the measurements were made in a direction perpendicular to the axis of the cone, and thus along the layers of material applied during the additive manufacturing process. Measurement through the layers was not possible due to the too-short length of the measurement section. Three measurements were made for each of the conical surfaces of a given type of valve. Each measurement within a given valve was in a direction parallel to the previous one.

### 2.5. Check Valve Testing Procedures

All check valves were tested in terms of internal leakage and opening pressure. The tests were carried out based on the recommendations of the ISO 6403:1988 standard [23]. The first test applied to cracking pressure. The testing rig consisted of a hydraulic powerpack with a fixed displacement pump, a pressure relief valve, two throttle valves (one of them serving as a shut-off valve), the tested valve, pressure sensors, and a flowmeter. A configuration of the rig is presented in Figure 5. The parameters of the test rig are presented in Table 2.

For the leakage tests, the configuration of the testing rig was similar to the previous test; the difference was the direction of the check valve—it was placed in the opposite direction so the flow would be forbidden. The modified version of the test rig is shown in Figure 6. 

## 3. Results and Discussion

Before the performance check, weight measurements of all AM valves were performed. The results were compared with the values measured for an originally manufactured counterpart. The results are shown in Figure 7. Additive manufacturing makes it possible to significantly reduce the weight of an element, which thus increases the economic and energy efficiency of the manufacturing process. This position has also been confirmed by other authors who have considered using AM for the production of parts for hydraulic systems [25,26].

During the tests, the opening pressure of each valve was examined. The results are presented in Figure 8 for the original, A, B, C, D, and E valves, respectively. 

For the originally manufactured valve, the opening pressure observed during the test was about 0.08 MPa, despite the fact that, according to the product data sheet, it was meant to be about 0.05 MPa. This difference is connected to the structure of the system. Due to the presence of hose after the examined valve, there is additional resistance in the system. The opening curve is clear, so it is easy to register the opening pressure. For the A valve, the opening pressure is comparable to the manufacturer’s valve and is about 0.09 MPa. 

For valves B, C, and E, it is difficult to clearly indicate the opening pressure value. This phenomenon may be connected with the structure of the seat, which is analyzed in a further section of the paper. For the D valve, the opening pressure is significantly higher than for the manufactured valve and is about 0.15 MPa. This may be the result of resistance in the outlet channel. In the other cases, the opening pressure is about 0.05 MPa.

The results of the leakage tests are presented in Table 3. Leakage should not exceed 1 cc/min, and the manufactured valve meets this requirement. For all prototypes, leakage is higher than required. A strong connection between the way the valve was prepared and the level of leakage was observed. For valves with postprocessed internal dimensions and threads, the leakage value was slightly higher than expected. For valves without postprocessing, the leakage was equal to the displacement of the pump, and so the main function of the valve was not executed.

Photos of the individual check valve seats that work with the poppet are shown in Figure 9. Valve seats that have not been finished are characterized by the presence of powder particles that have been melted and sintered to the surface during the SLM manufacturing process. This has a negative effect on the value of the surface roughness parameters and, in practice, on the accuracy of the adherence of the interacting surfaces. In addition, some of the sintered powder grains may detach from the surface and enter the hydraulic system under the influence of flowing oil, causing damage to other components. The presence of particles on the external and internal surfaces of additively manufactured components is a known effect in the literature and largely depends on the manufacturing parameters [27]. However, in the case of the considered issue, this phenomenon is intensified under the influence of the orientation of the manufactured element relative to the working platform. The layered nature of the production process using the SLM method causes the production of inclined surfaces to take place in a stepped manner and creates the so-called “staircase effect” [28,29]. This has a negative effect on the dimensional accuracy of the component and its surface roughness. 

In addition, the machined valve shown in Figure 9d has visible defects on the conical surface of the valve seat (marked with red squares). This is an effect related to the opening of subsurface pores during finishing. These voids may contain loose powder particles that, when in contact with the oil, may be flushed out and enter the hydraulic system. This effect can be minimized by optimizing the production parameters, with particular emphasis placed on the parameters responsible for scanning the outer shell in order to reduce or eliminate defects in the form of porosity. Due to the use of a forming tap to create threads, several defects were created in the machined area. Figure 9b,c,e show material particles that were formed as a result of kneading and partially moving the material in the direction of threading. Strong deformation of the material resulted in the formation of “growths”, as shown in Figure 9b,c. However, we see in Figure 9e that, as a result of strain hardening (and thus an increase in the hardness of the material and possible structural defects inside), a crack was formed. Due to the creation of this type of imperfection, it is clear that cutting taps should be used for the postprocess machining of this type of part.

The results of the leakage measurements and the valve opening pressure have points of reference in the roughness measurements and the accuracy of the valve seats. The graph presented in Figure 10 shows the average values of the Ra parameter for individual check valves. The conventionally manufactured valve, which met the requirements of ISO 6403:1988 in terms of leakage values, was characterized by the lowest value of the Ra parameter. Valve C was characterized by a similar value of Ra registered in relation to the conventionally manufactured valve. In the cases of the other valves, values over 10 times higher were recorded, which result from the specificity of the SLM process and the phenomena described in previous paragraphs. A possible reduction in roughness is possible through changing the orientation of the manufactured parts, as demonstrated in the work of Vishwakarma et al. [30], or through changing the parameters for producing contour paths [31]. High roughness worsens the contact of the poppet to the valve seat and thus reduces the tightness of the connection. This is confirmed by the leak result values for valves A, B, D, and E.

As for the difference in valve opening pressure, in addition to roughness, the angle of tilt of the seat cone of the valve may also have an influence. Figure 11a shows the angle of inclination of the cone of valve seat C. Due to the use of different tools than in the case of the production of the originally manufactured valve, this angle differs by 2.71°. Increasing this value results in the elongation of the generatrix of the cone and thus a change in the position of the poppet, despite there being the same spring setting in each tested valve.

The last issue concerns the shape of the edge of the valve seats, selected images of which are shown in Figure 11b–e. In the case of the original valve seat, the transition edge is rounded, which increases the contact area with the poppet and thus improves the tightness. In the case of the edge of valve seat C shown in Figure 11c, the rounding has not been taken into account, which results in only a linear connection with the poppet and a low tightness. In the case of the edges visible in Figure 11d,e, irregularities characteristic of parts produced via SLM without additional postprocessing are visible. However, the edge visible in Figure 11d is characterized by a smooth transition (valve B) in relation to the edge in Figure 11e (valve E). Despite the high roughness in both cases, valve B was characterized by lower leakage than valve E. This proves the importance of edge rounding in the originally manufactured valve.

## 4. Conclusions

AM technologies create possibilities for the production of hydraulic systems parts that are characterized by significantly reduced weight (about 91%), which is especially important in terms of energy consumption and fuel use. Such a mass reduction extends the possibility of using hydraulic systems in UAVs. However, the conducted research revealed a crucial disadvantage of using AM technology that needs to be subjected to additional postprocessing, including a complete subtractive manufacturing of internal threads. It was observed that using PBF–LB/M technology significantly affects hydraulic component parameters, especially the internal leakage values. The registered imperfections were mostly related to the increased surface roughness of the valve seats. 

The conducted research allowed us to draw the following conclusions:Using PBF–LB/M technology allowed for a reduction in the total weight of the produced valves by 91% (from 216.76 g to 18.67 g). Due to the fact that both valves were characterized by the very poor leakage results, it is crucial to use additional posttreatment techniques (especially for threads).In the case of opening pressure measurements, the type of production technology does not strongly affect this parameter.Leakage tests indicated significant differences between conventionally manufactured and AM valves. In all as-built AM parts, the internal leakage exceeds the standard requirements mainly because of increased surface roughness. The other factor that could affect the growth in internal leakage is a reduction in contact area caused by differences in valve seat shapes.According to the leakage test results, it is clearly visible that PBF–LB/M, as-built check valves indicate significant leakages that exceed a few orders of magnitude (parts A, B, D, and E). Such parts are disqualified from use in hydraulic applications. Only one AM-made check valve (C), after additional machining, was close to passing the required standard regulation (leakage lower than 1 cc/min). This kind of part could be used, i.e., in emergency situations during a lack of availability of spare parts or BDR.AM-ed check valves could be used in direct application only after very precise machining, in particular, the conical surface inside the check valve and the transition edge (described in Figure 11).

For future research, it is crucial to select the proper postprocessing technology that would allow for the improvement of the internal surfaces of AM valves. It seems that abrasive flow machining would be a promising postprocessing method for that kind of role. 

## Figures and Tables

**Figure 1 materials-16-07302-f001:**
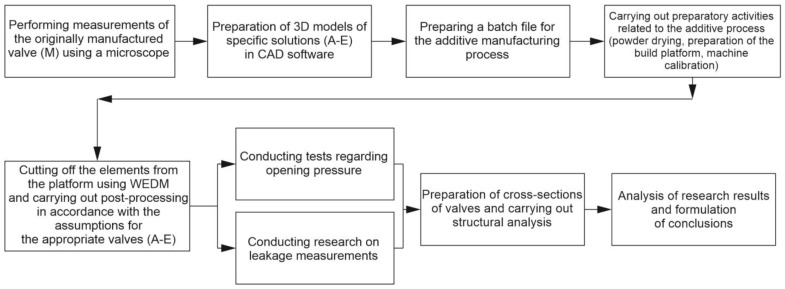
Flowchart of research procedure.

**Figure 2 materials-16-07302-f002:**
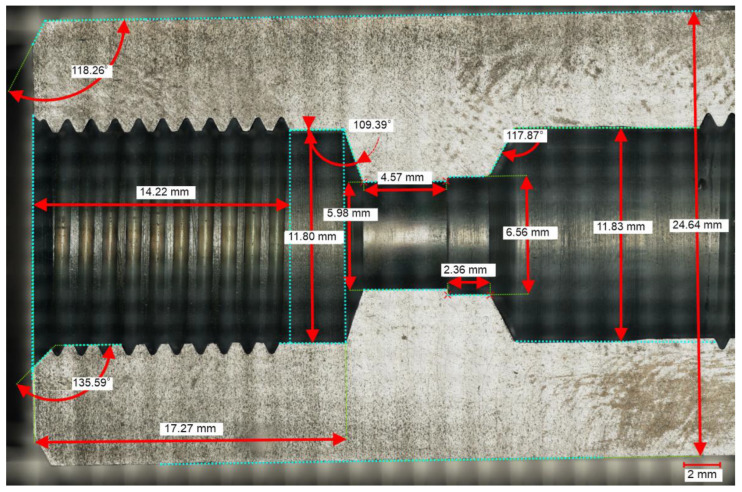
Cross-section of commercial check valve with dimensions.

**Figure 3 materials-16-07302-f003:**
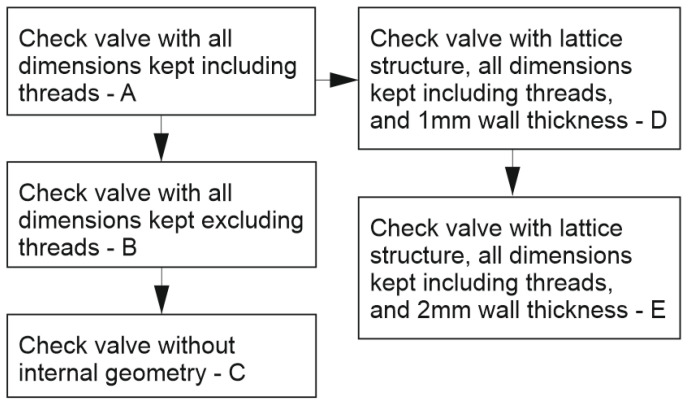
Structural diagram with all iterations of check valve geometries.

**Figure 4 materials-16-07302-f004:**
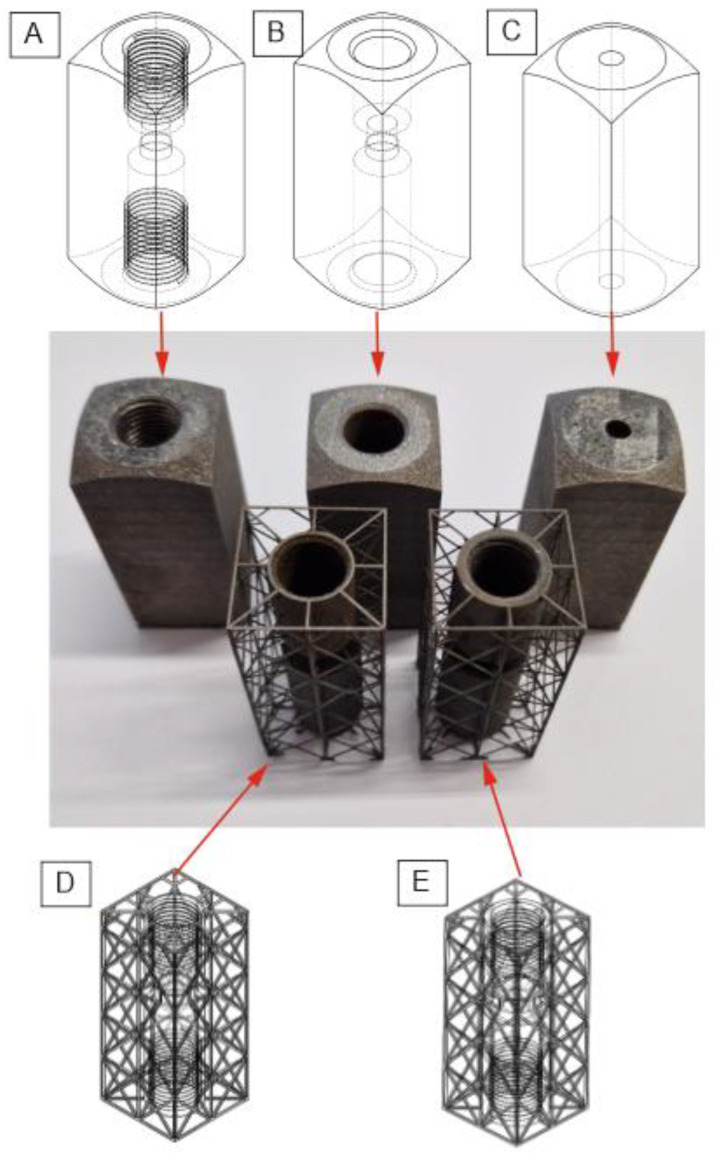
Five iterations of the AM check valve after the production process: (**A**) Check valve with all dimensions kept (including threads). (**B**) Check valve with all dimensions kept (excluding threads—for additional postprocessing of threads). (**C**) Check valve with external dimensions kept (for additional postprocessing of internal dimensions and threads). (**D**) Check valve with all dimensions kept (including threads), lattice structure, and 1 mm of wall thickness. (**E**) Check valve with all dimensions kept (including threads), lattice structure, and 2 mm of wall thickness.

**Figure 5 materials-16-07302-f005:**
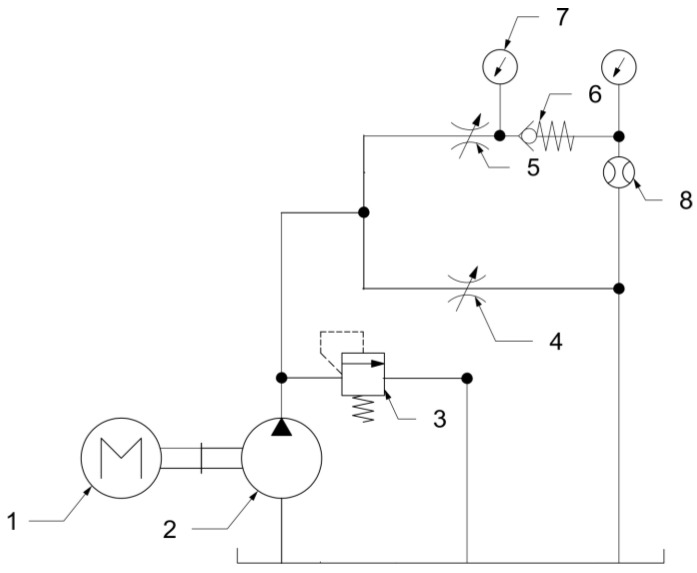
Testing rig for opening pressure tests: 1—motor; 2—hydraulic pump; 3—pressure relief valve; 4—throttle valve; 5—throttle valve; 6—tested check valve; 7—pressure gauge; 8—flowmeter.

**Figure 6 materials-16-07302-f006:**
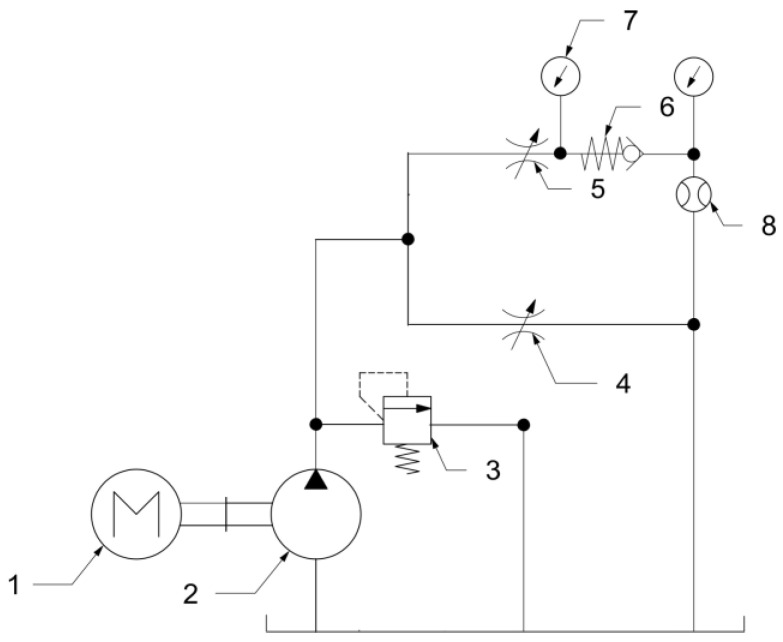
Testing rig for leakage tests: 1—motor; 2—hydraulic pump; 3—pressure relief valve; 4—throttle valve; 5—throttle valve; 6—tested check valve; 7—pressure gauge; 8—flowmeter.

**Figure 7 materials-16-07302-f007:**
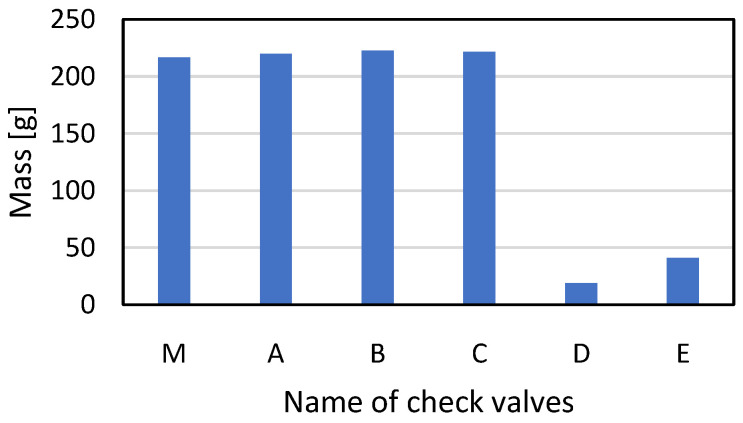
Total mass measurements of the originally manufactured (M) and AM valves (A–E).

**Figure 8 materials-16-07302-f008:**
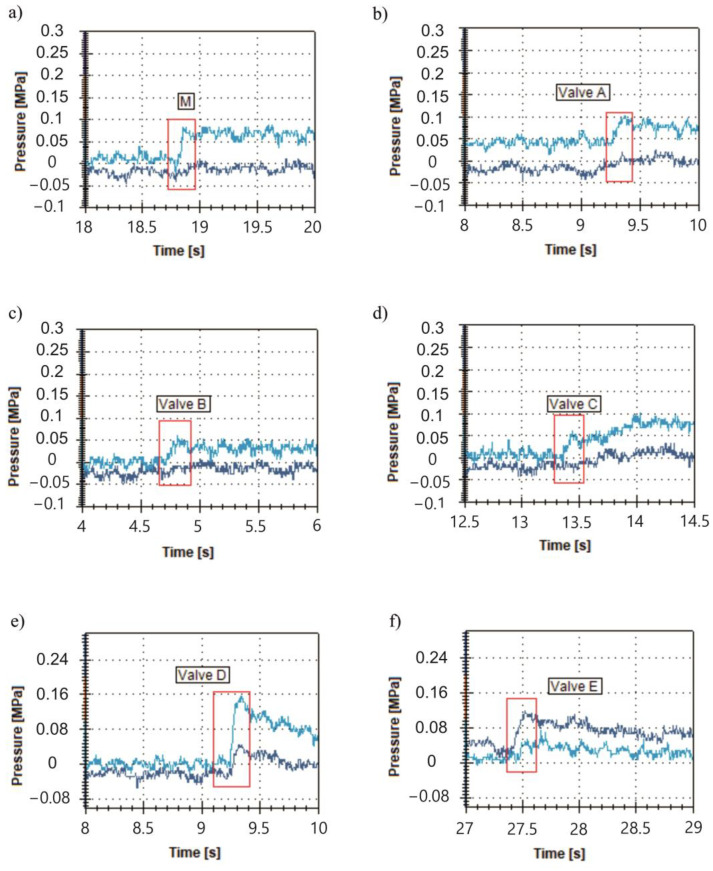
Opening pressure test results for valves: (**a**) manufactured one; (**b**) valve with all dimensions kept (including threads); (**c**) valve with all dimensions kept (excluding threads); (**d**) valve with external dimensions kept; (**e**) valve with all dimensions kept, lattice structure, and 1 mm of wall thickness; (**f**) valve with all dimensions kept, lattice structure, and 2 mm of wall thickness. Bright blue line—an oil pressure in the valve’s input; dark blue line—an oil pressure in the valve’s output.

**Figure 9 materials-16-07302-f009:**
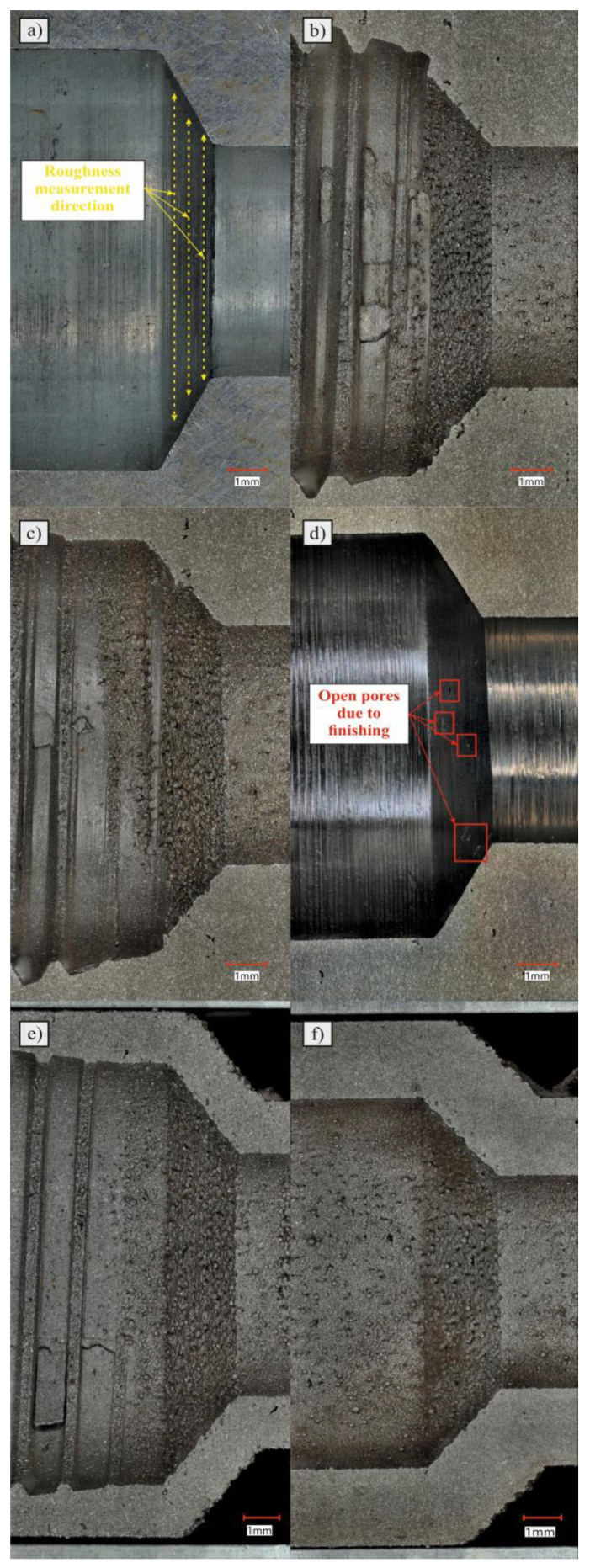
Valve seat of: (**a**) manufactured one; (**b**) valve with all dimensions kept (including threads); (**c**) valve with all dimensions kept (excluding threads); (**d**) valve with external dimensions kept; (**e**) valve with all dimensions kept, lattice structure, and 1 mm of wall thickness; (**f**) valve with all dimensions kept, lattice structure, and 2 mm of wall thickness.

**Figure 10 materials-16-07302-f010:**
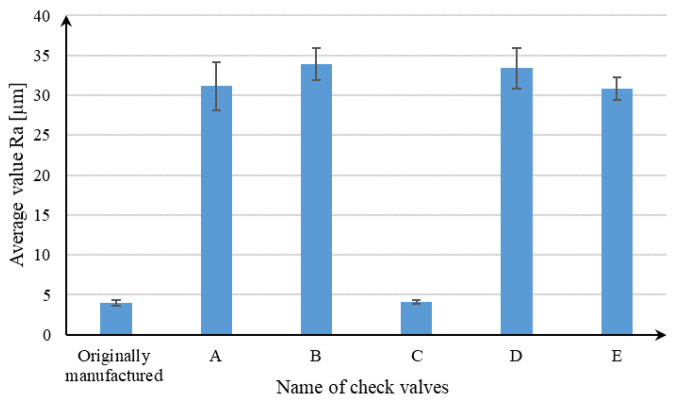
The average value of Ra parameters for individual check valves.

**Figure 11 materials-16-07302-f011:**
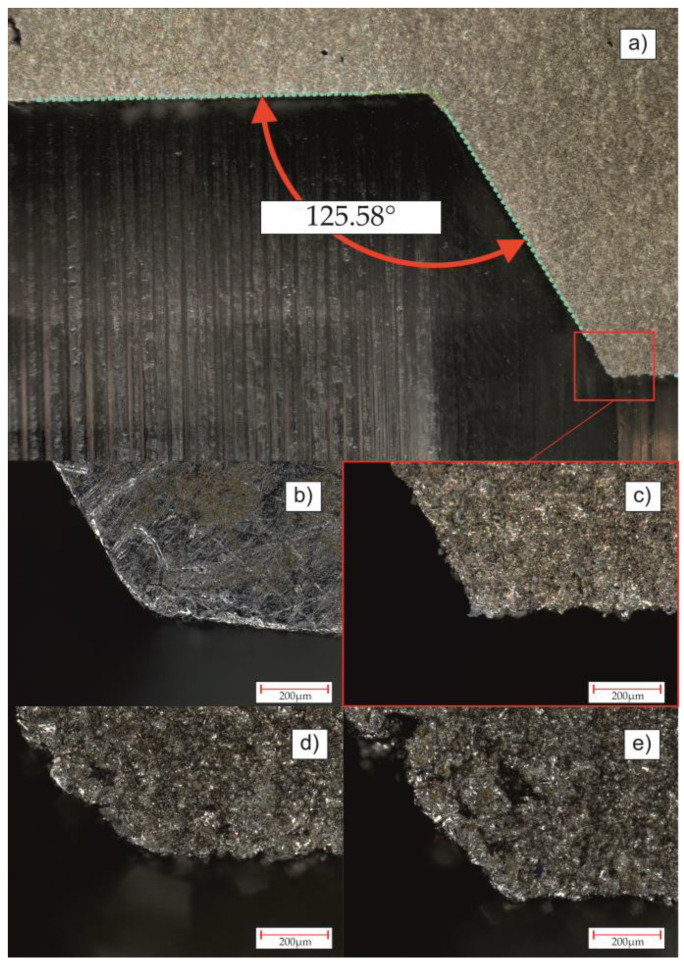
Microscopical images of transition edges of each valve: (**a**) part of valve C with visible measurement of the angle of inclination of the generatrix of the cone, transition edge of the valve seat; (**b**) originally manufactured; (**c**) C; (**d**) B; (**e**) E.

**Table 1 materials-16-07302-t001:** Process parameters used for PBF–LB/M of samples.

Parameter	Laser Power [W]	Exposure Velocity [mm/s]	Hatch Spacing [mm]	Layer Thickness [mm]	Energy Density [J/mm^3^]	Rotation Angle of Scanning Lines	Scanning Strategy
Value	175.5	750	0.12	0.03	65	67°	Stripes

**Table 2 materials-16-07302-t002:** Test rig parameters.

Parameter	Value
Maximum operating pressure [MPa]	16
Pump displacement [cc/rev]	1.3
Motor speed [rpm]	3000

**Table 3 materials-16-07302-t003:** Leakage test results.

Valve Type	M	A	B	C	D	E
Flow [cc/min]	0.9	3900	733	4.9	3900	3900

## Data Availability

Data are contained within the article.

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
