# Peer review of "Performance Analysis of Additively Manufactured Hydraulic Check Valves with Different Postprocessing"

_materials, 2023, doi:10.3390/ma16237302_

Round 1

Reviewer 1 Report

Comments and Suggestions for Authors

Based on the potential for additive manufacturing (AM) technology to produce lighter-weight components at a faster rate, this study investigates the feasibility of replacing traditional manufacturing hydraulic check valves with AM counterparts. Specifically, the authors utilized AM technology to manufacture five samples with varying degrees of processing, some of which were structurally complete while others required additional post-processing. Through testing the samples and the prototype's opening pressure and leakage pressure, several valuable conclusions were drawn. I believe this article demonstrates some degree of innovation and research value, but overall, it is rather rough, with numerous issues in writing and images that must be addressed.

1. In line 49, there should be a period "." before "Thus";

2. When introducing five types of work-pieces (Lines 117-126), I suggest adding corresponding structural diagrams for them to aid the readers understanding;

3. In Fig. 2, the codes A through E should be used to designate the five as-built work-pieces due to the author has named them in lines 117-126;

4. Lines 153-159 have been repeated in lines 117-126 above and in the annotation of Fig. 2. Therefore, heres should be removed;

5. In Fig. 3, the annotation "7" is not fully displayed in the right way. Please modify;

6. Fig. 6 has serious TYPOGRAPHICAL ERRORS and missing images. In particular, the important test curves of the C valve are missing. Please correct them;

7. The author said "For the originally manufactured valve opening pressure observed during the test was about 0.1 MPa, despite the fact that according to the product data sheet, it should be about 0.05 MPa.", how to explain this deviation (or a failure)? If the original manufactured valve, as a baseline for other AM valves, is a failure, the comparative tests will be meaningless;

8. The author said, " For the A valve opening pressure is lower than in the manufacturer's valve and is about 0.05 MPa". However, the curve value in Figure (b) seems to reach 0.1 MPa and then oscillate around 0.08 MPa, not as stated by the author of 0.05 MPa. What criteria does the author use to define the final opening pressure?

9. I observe some thread-like patterns on the inner sides of the workpieces in Fig. 7 b, c, and e. Are they caused by the so-called "staircase effect", or just thread? Why do such patterns not exist in Fig. 7(f)? The author should add explanations;

10. In the Conclusions (line 310), the author maintains that PBF-LB/M technology can reduce the total mass of the AM valves, but such valves simply cannot achieve usable performance after the authors testing (like the valves D and E). As shown in Table 5, the quality of the best C valve in this article is even slightly higher than that of the original valve. Therefore, I do not think the experimental and testing results can support this conclusion.

Author Response

Dear Reviewer,

On behalf of all authors, I would like to thank you for taking the time to read our manuscript and put in your comments which allowed us to improve the quality of our work. Below you can find our answers related to each of your comments.

  1. In line 49, there should be a period "." before "Thus";

Ad. 1. We are sorry for this editorial mistake. A period has been added.

  1. When introducing five types of work-pieces (Lines 117-126), I suggest adding corresponding structural diagrams for them to aid the reader’s understanding;

Ad.2. Thank you very much for your advice. We put an additional structural diagram just below the lines you mentioned.

  1. In Fig. 2, the codes A through E should be used to designate the five as-built work-pieces due to the author has named them in lines 117-126;

Ad.3. We are thankful for pointing out this issue. It has been corrected.

  1. Lines 153-159 have been repeated in lines 117-126 above and in the annotation of Fig. 2. Therefore, here’s should be removed;

Ad. 4. The repeated lines have been removed.

  1. In Fig. 3, the annotation "7" is not fully displayed in the right way. Please modify;

Ad. 5. Figure has been modified.

  1. Fig. 6 has serious TYPOGRAPHICAL ERRORS and missing images. In particular, the important test curves of the C valve are missing. Please correct them;

Ad. 6. We are extremely sorry for this editorial mistake, it has to be connected with the submission process, some parts were damaged. Now we have done our best to avoid such an issue and modified the figure to ensure proper visibility.

  1. The author said "For the originally manufactured valve opening pressure observed during the test was about 0.1 MPa, despite the fact that according to the product data sheet, it should be about 0.05 MPa.", how to explain this deviation (or a failure)? If the original manufactured valve, as a baseline for other AM valves, is a failure, the comparative tests will be meaningless;

Ad. 7. Thank you very much for this valuable comment. First of all, we should be more precise – according to our tests, the opening pressure for the originally manufactured valve is 0.08 MPa. Secondly, this difference is connected to the structure of the system – due to the presence of a hose after the examined valve, there is additional resistance in the system. We added a proper explanation in the manuscript:

“This difference is connected to the structure of the system – due to the presence of hose after the examined valve there is additional resistance in the system.”

  1. The author said, " For the A valve opening pressure is lower than in the manufacturer's valve and is about 0.05 MPa". However, the curve value in Figure (b) seems to reach 0.1 MPa and then oscillate around 0.08 MPa, not as stated by the author of 0.05 MPa. What criteria does the author use to define the final opening pressure?

Ad. 8. Thank you for this comment. We are extremely sorry for this mistake. The description put in the article applies to B valve obviously. The description for A valve has been corrected:

" For the A valve opening pressure is comparable to  the manufacturer's valve and is about 0.09 MPa”

To determine the opening pressure of the valve the procedure was as follows:

The pump was powered, and the throttle valve 4 was fully open. There was no flow behind the tested check valve. Then the throttle valve was closed until there was a flow of about 5 cc/min. The procedure was prepared based on the ISO standard.

  1. I observe some thread-like patterns on the inner sides of the workpieces in Fig. 7 b, c, and e. Are they caused by the so-called "staircase effect", or just thread? Why do such patterns not exist in Fig. 7(f)? The author should add explanations;

Ad 9. Thank you very much for your comment. Images 7b, c, and e show defects resulting from the method of threading. A forming tap was used for this process. The deformation of the material and its movement along the axis of the cylinder causes the so-called "growths", and deformation strengthening may even lead to cracks such as in photo 7e. This explanation has been added to the article.

  1. In the Conclusions (line 310), the author maintains that PBF-LB/M technology can reduce the total mass of the AM valves, but such valves simply cannot achieve usable performance after the author’s testing (like the valves D and E). As shown in Table 5, the quality of the best C valve in this article is even slightly higher than that of the original valve. Therefore, I do not think the experimental and testing results can support this conclusion.

Ad.10. Thank you for pointing out this issue. We were not precise enough in this conclusion. Both of the lattice structured check valves did not have a post-treatment of threads included. Such a solution is possible to use but with additional postprocessing of internal threads. We rephrased this conclusion, and now it has the following form:

“Using PBF – LB/M technology allowed a reduction in the total weight of the produced valves by 91% (from 216.76 g to 18.67 g). Due to the fact that both valves were characterized by the worst leakage results, it is crucial to use additional post-treatment techniques - especially for internal threads.”

Reviewer 2 Report

Comments and Suggestions for Authors

The topic of the manuscript is really interesting and quite well done. However, there are some shortcomings that need to be removed.

1) The description of the samples is repeated several times (lines 117-126, Fig. 2 caption in lines 146-152 and again in the text in lines 153-159). At least the last repeating is superfluous and can be removed.

2) It is necessary to correct the presentation of graphs in Fig. 6. Graphs c) and e) are moved below caption of the Table 3, graph d) is covered by graph f). Therefore, this part of the manuscript can be hard to understand and some information (graph d) is lost.

3) There are some small problems with notation of the text (namely in the Introduction): line 47 - greater space before "Such"; line 49 - missing space before "[6]" and dot after it; missing space before "Based" in line 95.

Comments on the Quality of English Language

There are no important deficiencies in language.

Author Response

Dear Reviewer,

On behalf of all authors, I would like to thank you for taking the time to read our manuscript and put in your comments which allowed us to improve the quality of our work. Below you can find our answers related to each of your comments.

1) The description of the samples is repeated several times (lines 117-126, Fig. 2 caption in lines 146-152 and again in the text in lines 153-159). At least the last repeating is superfluous and can be removed.

Ad. 1. The repeated lines (153-159) have been removed.

2) It is necessary to correct the presentation of graphs in Fig. 6. Graphs c) and e) are moved below caption of the Table 3, graph d) is covered by graph f). Therefore, this part of the manuscript can be hard to understand and some information (graph d) is lost.

Ad. 2. We are extremely sorry for this editorial mistake, it has to be connected with the submission process, some parts were damaged. Now we have done our best to avoid such an issue and modified the figure to ensure proper visibility.

3) There are some small problems with notation of the text (namely in the Introduction): line 47 - greater space before "Such"; line 49 - missing space before "[6]" and dot after it; missing space before "Based" in line 95.

Ad. 3. Thank you for your vigilance. The editorial mistakes have been fixed.

Reviewer 3 Report

Comments and Suggestions for Authors

This paper studies the possibility of obtaining hydraulic control valves by selective laser sintering technology.

The conclusions of the experiments are that the application of additive manufacturing technology is not sufficient but requires post-processing of the obtained products.

In my opinion, to be considered relevant, the experiments should be repeated a sufficient number of times to allow statistical processing.

As far as can be understood from the paper, for each of the proposed examples only one copy was made which does not guarantee the repeatability of the results.

From a typing point of view, the legend of figure 2 seems to repeat part of the legend of figure 1.

Figures 6 c, e, and f are offset which makes it difficult to interpret the results.

Author Response

Dear Reviewer,

On behalf of all authors, I would like to thank you for taking the time to read our manuscript and put in your comments which allowed us to improve the quality of our work. Below you can find our answers related to each of your comments.

1.The conclusions of the experiments are that the application of additive manufacturing technology is not sufficient but requires post-processing of the obtained products.

Ad.1. That was not a 100% message that we wanted to provide in the conclusion part. We have rephrased it: “However, the conducted research revealed a crucial cons of using AM technology that needs to be subjected to additional postprocessing, including a complete, subtractive manufacturing of internal threads.”

So, we mean that in the case of hydraulic parts production, it is necessary to use postprocessing but we tried to answer the question – how advanced postprocessing we should provide to obtain proper performance properties of hydraulic check valves.

  1. In my opinion, to be considered relevant, the experiments should be repeated a sufficient number of times to allow statistical processing.

Ad. 2. Thank you for this comment. We should mention that each test was carried out five times and the representative result has been presented in our manuscript. Hydraulic systems are considered to be stochastic systems, this is why deeper statistical analysis was not used in that kind of research.

  1. As far as can be understood from the paper, for each of the proposed examples only one copy was made which does not guarantee the repeatability of the results.

The aim of our research was to analyze the impact of post-processing on specific properties of valves determined by their intended use. We assumed that the AM of each check valve’s geometry lacks process issues and it is sufficient to produce one piece of each valve to check whether certain post-process activities would be sufficient to ensure full functionality of the valve. We used well-known methods such as sandblasting, threading, and mechanical machining, and checking the repeatability of these processes is widely recognized. However, the repeatability of the additive manufacturing process is a completely separate topic that was taken into account in other research papers.

  1. From a typing point of view, the legend of figure 2 seems to repeat part of the legend of figure 1.

This part was also taken up by another reviewer. When we were doing a correction based on another reviewer's comment, we took your opinion into account to fix the mentioned issue. Thank you.

Ad. 4. Thank you for your comment. This repetition was made on purpose to provide faster recognition of each valve in Figure 2.

  1. Figures 6 c, e, and f are offset which makes it difficult to interpret the results.

Ad. 5. We are extremely sorry for this editorial mistake, it has to be connected with the submission process. We modified the figure to ensure proper visibility.

Reviewer 4 Report

Comments and Suggestions for Authors

The article needs some improvements to consider as a publication in the journal. In this way, more specific comments are described in the comments to the authors as follows.

+ The title can be improved as: Performance analysis of the additively manufactured hydraulic check valves with different postprocessing

Last paragraph of the Introduction should explicitly explain the gap of the studies reviewed above and the contribution of this manuscript.

+ In materials and methods section, can be useful a flowchart of the process developed to AM.

+ Please, avoid the use of personal pronouns. (own scanning electron microscopy (SEM)). In this way, the equipments and test need to be cleary detailed, like configurations and others.

+ Why the authors use spherical form (with diameters ranging from 20 105 to 63 μm) and these diameters?

+ What is the objective to prepare five types of workpieces during the design process? Then, can be cool add the designs developed in SW software.

+ I don't understand, if the authors use SEM or optical microscopy for the tool figures. 

+ The authors can mention some wear predominate mechanisms or defects in manufacturing?

+ In general, the discussion policy is weak and insufficient in terms of academic manner. The authors should present more detail with novel achievements to better draw readers' attention.

+ The article shows coherent has results, concise, and concrete. Likewise, the discussion must be confronted with other authors; then, you must include references for discussion. Compare with other similar research and relate the percentage of improvement and thus the aim to develop this type of material for brake disc application.

Comments on the Quality of English Language

Minor issues

Author Response

Dear Reviewer,

On behalf of all authors, I would like to thank you for taking the time to read our manuscript and put in your comments which allowed us to improve the quality of our work. Below you can find our answers related to each of your comments.

  1. The title can be improved as: Performance analysis of the additively manufactured hydraulic check valves with different postprocessing

Ad. 1. Thank you for this opinion. The title has been improved according to your suggestion.

  1. Last paragraph of the Introduction should explicitly explain the gap of the studies reviewed above and the contribution of this manuscript.

Ad.2. Thank you very much for this comment. We extended the last sentence of the introduction that would better highlight the research gap which we wanted to fill. This sentence has now the following form:
“Such an analysis would be a helpful tool for DfAM of that kind of part and answer the very important question – how advanced postprocessing should be provided to obtain proper performance properties of hydraulic check valves?”

3.In materials and methods section, can be useful a flowchart of the process developed to AM.

Ad 3.  The flowchart has been added to the Material and Methods chapter in section 2.1 Research procedure.

  1. Please, avoid the use of personal pronouns. (own scanning electron microscopy (SEM)). In this way, the equipments and test need to be cleary detailed, like configurations and others.

Ad.4. We are sorry for this issue. In the original version of manuscript we put a SEM image of the powder, but we decided to remove it but we did not remove it from the text. It has been properly fixed.

  1. Why the authors use spherical form (with diameters ranging from 20 105 to 63 μm) and these diameters?

Ad.5. It is a typical shape of powder particles dedicated to SLM technology that was used in our research. A range between 20 and 63 μm allows for maintaining proper flowability of the powder and avoiding the generation of powder particles agglomerating. The powder in such condition has been delivered from the supplier - (Carpenter Additive, Philadelphia, PA, USA).

6.What is the objective to prepare five types of workpieces during the design process? Then, can be cool add the designs developed in SW software.

Ad.6. Our objective was to answer the very important question – how advanced postprocessing should be provided to obtain proper performance properties of hydraulic check valves? - we put such a statement in the last sentence of the introduction. Based on your advice we put SW screenshots in figure 3. We decided to put the images with internal geometry visible to better clarify our designs.

7.I don't understand, if the authors use SEM or optical microscopy for the tool figures.

Ad. 7. As we mentioned earlier (comment 4), an information about the SEM image was provided by a mistake. It has been removed.

  1. The authors can mention some wear predominate mechanisms or defects in manufacturing?

Defects resulting from manufacturing, not only the printing process, but also post-processing, have already been described in the article:

“Images of individual check valve seats that work with the poppet are shown in Fig. 7. Valve seats that have not been finished are characterized by the presence of powder particles that have been melted and sintered to the surface during the SLM manufacturing process. This has a negative effect on the value of surface roughness parameters, and in practice on the accuracy of adherence of the interacting surfaces. In addition, some of the sintered powder grains may detach from the surface and enter the hydraulic system under the influence of flowing oil causing damage to other components. The presence of particles on the external and internal surfaces of additively manufactured components is a known effect in the literature and largely depends on the manufacturing parameters  [24] . However, in the case of the considered issue, this phenomenon is intensified under the influence of the orientation of the manufactured element relative to the working platform. The layered nature of the production process using the SLM method causes the production of inclined surfaces to take place in a stepped manner and creates the so-called “staircase effect”  [25,26] . This has a negative effect on the dimensional accuracy of the component and its surface roughness. 

In addition, the machined valve shown in Fig. 7d has defects visible on the conical surface of the valve seat (marked with red squares). This is an effect related to the opening of subsurface pores during finishing. These voids may contain loose powder particles which, when in contact with the oil, may be flushed out and enter the hydraulic system. This effect can be minimized by optimizing the production parameters, with particular emphasis on the parameters responsible for scanning the outer shell in order to reduce or eliminate defects in the form of porosity. Due to the use of a forming tap to create threads, several defects were created in the machined area. Figures 7b, c, e show material particles that were formed as a result of kneading and partially moving the material in the direction of threading. Strong deformation of the material resulted in the formation of "growths" as in Figures 7b and 7c. However, in Figure 7e, as a result of strain hardening, and thus an increase in the hardness of the material and possible structural defects inside, a crack was formed. Due to the creation of this type of imperfections, it is clear that cutting taps should be used for post-process machining of this type of parts.”

It is difficult for us to describe the process of valve wear because their interior was not analyzed before testing related to opening pressure and leaks.

9.In general, the discussion policy is weak and insufficient in terms of academic manner. The authors should present more detail with novel achievements to better draw readers' attention.

Unfortunately, we cannot fully agree with this comment. We tried to discuss all possible phenomena that occurred in our case studies. We tried to search for something similar in the literature and put proper citations in the manuscript. Available literature shows in most cases how process parameters affect surface roughness but none of them discuss how this parameter affects performance properties of hydraulic parts. We agree with the reviewer, that the discussion is not as significant as it is in more common topics about AM technologies, but it is not caused by our ignorance but by a low amount of literature related to our topic. We hope that you will understand our point.

10. The article shows coherent has results, concise, and concrete. Likewise, the discussion must be confronted with other authors; then, you must include references for discussion. Compare with other similar research and relate the percentage of improvement and thus the aim to develop this type of material for brake disc application.

Thank you very much for drawing attention to this aspect. In the discussion section, we referred to additional research conducted in a similar field as follows:

“Additive manufacturing makes it possible to significantly reduce the weight of an element and thus increases the economic and energy efficiency of the manufacturing process. This position was also confirmed by other authors who considered the production of parts of hydraulic systems using AM [x1, x2]”

“A possible reduction in roughness is possible by changing the orientation of the manufactured parts, as demonstrated in the work of Vishwakarma et al. [x3] or by changing the parameters for producing contour paths [x4]”

24. A comparative life cycle assessment of a selective-laser-melting-produced hydraulic valve body using design for Property Yanan Wang , Tao Penga , Yi Zhu, Yang Yang , Renzhong Tanga

25. A Review of Hydraulic and Pneumatic Aggregates Manufacturing By Additive Technologies Smelov V.G.*, Kokareva V.V., Agapovichev A.V

26. Effect of build orientation on microstructure and tensile behaviour of selectively laser melted M300 maraging steel, Jaydeep Vishwakarma, K. Chattopadhyay, N.C. Santhi Srinivas

27. Wang, D.; Lv, J.; Wei, X.; Lu, D.; Chen, C. Study on Surface Roughness Improvement of Selective Laser Melted Ti6Al4V Alloy. Crystals 2023, 13, 306. https://doi.org/10.3390/cryst13020306

Reviewer 5 Report

Comments and Suggestions for Authors

The paper addresses the use of AM technology for production of hydraulic parts. The claims are well supported by the results, and I appreciate the candid admission that the authors presented related to the drawbacks of such a process. This work can potentially open up further research routes to optimize AM for hydraulic parts in the future. My only comment is that the paper needs a thorough proofreading before it is ready for publication; several minor grammatical and spelling errors throughout the manuscript.

Comments on the Quality of English Language

 Several minor grammatical and spelling errors throughout the manuscript.

Author Response

Dear Reviewer, we are very pleased by your kind words. We analyzed the whole paper and corrected all language issues.  

Round 2

Reviewer 1 Report

Comments and Suggestions for Authors

The author has made detailed revisions. I have no further comments and agree to publish. 

Reviewer 3 Report

Comments and Suggestions for Authors

I propose acceptance of the paper in its present form.